# A Fun-Guide to Innate Immune Responses to Fungal Infections

**DOI:** 10.3390/jof8080805

**Published:** 2022-07-29

**Authors:** Thomas B. Burgess, Alison M. Condliffe, Philip M. Elks

**Affiliations:** 1The Bateson Centre, University of Sheffield, Western Bank, Sheffield S10 2TN, UK; tburgess2@sheffield.ac.uk; 2Department of Infection, Immunity and Cardiovascular Disease, Medical School, University of Sheffield, Beech Hill, Sheffield S10 2RX, UK; a.m.condliffe@sheffield.ac.uk

**Keywords:** fungal infections, antifungal immunity, host–pathogen interaction, immune dysregulation, host-directed therapy

## Abstract

Immunocompromised individuals are at high risk of developing severe fungal infections with high mortality rates, while fungal pathogens pose little risk to most healthy people. Poor therapeutic outcomes and growing antifungal resistance pose further challenges for treatments. Identifying specific immunomodulatory mechanisms exploited by fungal pathogens is critical for our understanding of fungal diseases and development of new therapies. A gap currently exists between the large body of literature concerning the innate immune response to fungal infections and the potential manipulation of host immune responses to aid clearance of infection. This review considers the innate immune mechanisms the host deploys to prevent fungal infection and how these mechanisms fail in immunocompromised hosts. Three clinically relevant fungal pathogens (*Candida albicans*, *Cryptococcus* spp. and *Aspergillus* spp.) will be explored. This review will also examine potential mechanisms of targeting the host therapeutically to improve outcomes of fungal infection.

## 1. Introduction

Fungal infections are responsible for over 1.7 million deaths per year globally, roughly 200,000 more deaths than the bacterial disease tuberculosis [1], yet the immune responses to fungal infections are much less well studied and understood than those to bacterial infections. Because the burden of fungal diseases is greatest in tropical countries, fungal infections are frequently underfunded [2]. For example, it is estimated that for every individual that dies of bacterial meningococcal meningitis, USD 2458 is spent on research, while only USD 31 is spent per individual that dies of fungal cryptococcal meningitis, despite being responsible for 20 times the number of deaths [3]. Awareness within the general public is low, with less than a third of surveyed Americans being aware of fungal diseases [4].

The severity of fungal disease can range from minor, superficial infections (approximately 1 billion cases worldwide) to severe or life threatening conditions, such as chronic pulmonary aspergillosis (3 million cases worldwide) and severe invasive candidiasis (750,000 cases worldwide) [2]. Severe disease is most common in immunocompromised individuals [5,6]. Major risk factors include comorbidities with HIV/AIDS or tuberculosis, treatments for disease that requires immunomodulation, such as stem cell transplantation, or specific genetic defects resulting in primary immunodeficiency [2,7,8]. Inflammatory conditions such as chronic obstructive pulmonary disease and asthma also predispose patients to fungal infection, but with lower relative risk.

Fungal infections are most commonly treated with antifungal drugs, of which there are three main classes: polyenes, azoles and echinocandins [9]. Because fungi are eukaryotes, identification of specific antifungal targets that do not harm host cells is challenging. Development of new antifungals has been slow, with only 1 antifungal being approved in the last 10 years (isavuconazole, approved in 2015) [10] and so there is a shortage of new antifungal drugs [11,12]. Increasing antifungal resistance is a compounding issue. In one study of 54 patients across 3 continents, 93% of *Candida auris* patient isolates were resistant to fluconazole, with 41% resistant to 2 antifungal classes and 4% resistant to 3 classes [13]. Antifungal resistant infections are not only emergent, but are deadly. Azole-resistant *Aspergillus fumigatus* prevalence among patients in the Netherlands was 5.3%, but had an 88.0% fatality rate [14]. There is, therefore, a pressing need to develop new treatments for fungal infections.

Fungal spores are ubiquitous in the environment and are encountered on a daily basis [15]. Fungi that enter the body are usually controlled by our innate immune system, preventing disease from developing. Detection of fungal surface ligands by Pattern Recognition Receptors (PRRs) triggers a pro-inflammatory response, resulting in innate immune activation and elimination of fungi when the immune response is effective, or ineffective clearance and development of disease when immunity fails [16,17,18]. Focussing on 3 clinically relevant fungal pathogens (*Candida albicans, Cryptococcus neoformans* and *Aspergillus fumigatus*), this review aims to address the gap between our understanding of innate immunity to fungal infections and the development of novel host-directed strategies to combat infections in patients. The review will consider the role of the innate immune system in responding to fungal infections, how failures of innate immunity can result in severe fungal disease and how the innate immune system could be targeted therapeutically in novel treatments for fungal infections.

## 2. Innate Immune Control of Fungal Infections

### 2.1. Barriers to Fungal Entry

The first components of our protection against invading fungi are the physical and anatomical barriers that prevent entry of pathogens, primarily skin and mucosal membranes [17]. The skin is colonised by a range of commensal microorganisms, the main fungal species being members of the *Malassezia* genus [19,20]. Tight junctions in the epithelia form a physical barrier to fungal entry, while *C. albicans* colonisation on the skin has been shown to be controlled by skin-resident dendritic cells [17,21].

The primary mechanism of pulmonary exposure to fungi is inhalation of fungal spores, most commonly *Aspergillus* spp. [22]. Once in the respiratory tract, tight junctions between epithelial cells prevent fungal invasion into the host. A layer of mucus helps trap fungal spores, allowing cilia to move trapped fungi out of the respiratory tract to be coughed up or swallowed into the digestive tract to be destroyed by stomach acid [23].

### 2.2. Host Recognition of Fungi

Fungal pathogens can circumvent physical barriers and gain entry to the host in the case of a barrier break, e.g., injury. PRRs are able to detect a range of conserved structures on pathogens, known as pathogen associated molecular patterns (PAMPs), as well as detecting the damage caused by pathogens, known as damage associated molecular patterns (DAMPs) [24]. This triggers an intracellular signalling cascade, leading to production of effector proteins and recruitment of innate immune cells (Table 1). There has been a large effort to increase our understanding of PRRs involved in antifungal immunity. C-type lectin receptors (CLRs), such as Dectin-1 and Mincle, are PRRs that have been demonstrated to detect fungi. Dectin-1 specifically detects β-1,3-glucan, a fungal cell wall carbohydrate, stimulating NF-κB signalling, inflammasome activation, phagocytosis and production of reactive oxygen species (ROS) [25]. Double stranded RNA (a PAMP usually associated with viral infections) in *A. fumigatus* infection is detected by RIG-I-like receptors, which stimulates MDA5/MAVS signalling. Type III interferon expression is entirely reliant on MDA5/MAVS, whereas Type I interferon expression was also triggered through alternative mechanisms [26]. Type III interferon appears to be a critical regulator in neutrophil activation and antifungal immunity [27], suggesting MDA5/MAVS signalling is critical in the antifungal immune response.

### 2.3. Macrophages

Macrophages are a key innate immune cell type in fungal infection control. Recruitment of macrophages to sites of fungal infection is a highly dynamic process. Macrophages cluster around *A. fumigatus* infection in zebrafish and play a role in preventing the yeast-to-hyphae transition, which is associated with increased pathogenicity [38]. The number of macrophages in clusters was highly dynamic, though whether decreases in macrophage density was caused by reverse migration, apoptosis/pyroptosis or cell death was not revealed. In silico analysis of *Mucor circinelloides* infection, supported by observations in zebrafish models of infection, revealed the number of phagocytes present at the site of infection is critical to infection control [39]. The size of macrophage clusters may play a crucial role in control of fungal infections. Resident tissue macrophages have been demonstrated to congregate and “cloak” tissue microlesions with pseudopods in an in vivo mouse sterile injury model, which concealed pro-inflammatory debris, prevented neutrophil swarming and reduced collateral tissue damage caused by neutrophil-mediated inflammation [40]. Though not yet shown in a model of fungal infection, macrophage clustering may similarly facilitate cloaking of damage caused by fungi, limiting the pro-inflammatory neutrophil response and preventing excess tissue damage. Conversely, damage caused by fungi could be too large to effectively cloak, permitting the hyperinflammatory response and collateral tissue damage observed in alternative models of fungal infection [41].

Following recruitment, the primary mechanism of pathogen clearance by macrophages is phagocytosis. Macrophage PRRs or Fc receptors bind to fungal PAMPs or opsonising antibodies, respectively, triggering engulfment of the fungus [42]. Two mechanisms of fungal engulfment have been described: zipper phagocytosis and coiling phagocytosis (Figure 1) [43,44]. Phagocytosis of *C. neoformans* is typically facilitated by crosslinking of Fcγ receptors and *C. neoformans* bound IgG antibodies. Inability to form lipid rafts with closely localised Fcγ receptors prevents IgG mediated phagocytosis, though complement mediated phagocytosis is unaffected [45]. Following investigation in mouse knockout models, Anion Exchanger 2 has been suggested as a critical regulator of engulfment, through regulation of Dectin-1 expression, and fungal killing, by affecting intracellular pH homeostasis in macrophages [46]. Engulfed fungi are held in the phagosome, which fuses with the lysosome to form a phagolysosome. Acidification of the phagolysosome allows fungal degradation by acid-dependent proteases (such as Cathepsin D), combined with fungal killing by ROS and reactive nitrogen species (RNS) [42,47,48]. Hatinguais et al., demonstrated mitochondrial ROS, produced via reverse electron transport, not only contribute to the destruction of phagocytosed *A. fumigatus* conidia, but also trigger production of TNF-α and IL-1β in vitro, stimulating further antifungal responses [49]. Although inhibition of mitochondrial H_2_O_2_ impaired phagocytosis of *A. fumigatus* by alveolar macrophages in a mouse model, neutrophil activity was not impaired and survival and fungal burden were not affected [50]. Hence, while mitochondrial ROS are important in for the antifungal activity of alveolar macrophages, NADPH oxidase activity is able to compensate effectively, displaying redundancy in the host antifungal response. While lysosomal degradation is the typical outcome of phagocytosis, macrophage-to-macrophage transfer (termed “Dragotcytosis”) of *C. neoformans* has also been observed in vitro [51]. Shah et al., observed a similar phenomenon in *A. fumigatus* infection. Hyphal growth of phagocytosed *A. fumigatus* caused macrophage necrosis, triggering macrophage-to-macrophage transfer of germinating *A. fumigatus*, preventing fungal escape [52]. Unlike dragotcytosis, transfer of *A. fumigatus* was macrophage necrosis-dependent. However, the biological significance of these phenomena is unknown. Vomocytosis (also referred to as nonlytic exocytosis) is the expulsion of phagocytosed particles without degradation into the extracellular environment and occurs in macrophages in vitro during *C. albicans* and *C. neoformans* infection [53,54]. In mammalian in vitro and zebrafish in vivo models of cryptococcosis, vomocytosis is regulated by the MAP kinase ERK5. Viral infection and type I interferon signalling have been associated with enhanced rates of vomocytosis in vitro [55,56]. It is possible that the expulsion phase of dragotcytosis operates by a similar mechanism.

Phagocytosis of fungi is not always possible: *C. albicans* and *A. fumigatus* hyphae may become too long to phagocytose and *Cryptococcus neoformans* titan cells are too large to phagocytose [57,58,59]. Inability to phagocytose a pathogen typically leads to frustrated phagocytosis, a process in which there is downregulation of phagocytosis mechanisms and a strong inflammatory response mediated by IL-1β [60,61].

Phagocytosis of *C. albicans* can trigger a yeast-to-hyphae transition, leading to macrophage killing through mechanical piercing by hyphae or induction of pyroptosis, allowing escape of *C. albicans* [62]. To counteract this, macrophages are able to fold phagocytosed fungal hyphae at septal junctions (Figure 1), resulting in significantly reduced hyphal growth and disruption to the cell wall [63]. Exactly how much hyphal folding contributes to fungal clearance is unknown, however, this represents a previously uncharacterised macrophage function, which may be relevant to other hyphal pathogens.

Macrophages are a highly heterogenous population, existing on a spectrum of behaviours between M1, pro-inflammatory phenotypes and M2, wound healing phenotypes [64]. Proteomic analysis revealed a pro-inflammatory to wound healing phenotypic switch in *C. albicans* infection, whereas *C. neoformans* infection drives macrophages into a naïve M0 phenotype [65,66]. Stimulating an M1 phenotype led to decreased fungal burden and increased survival of mouse models in *C. albicans* and *C. neoformans* infection [67,68]. Promotion of an M2 phenotype in *A. fumigatus* infection reduces control of infection, corresponding with the other fungal pathogens [69]. This reveals a mechanism to avert pro-inflammatory macrophage polarisation to the detriment of the host, driven by immunosuppressive drugs (e.g., steroids) or interactions with fungal pathogens. It could be possible to improve outcomes of fungal infection by therapeutically promoting M1 macrophage polarisation, though this must be done with caution to prevent excess, harmful inflammation.

**Figure 1 jof-08-00805-f001:**
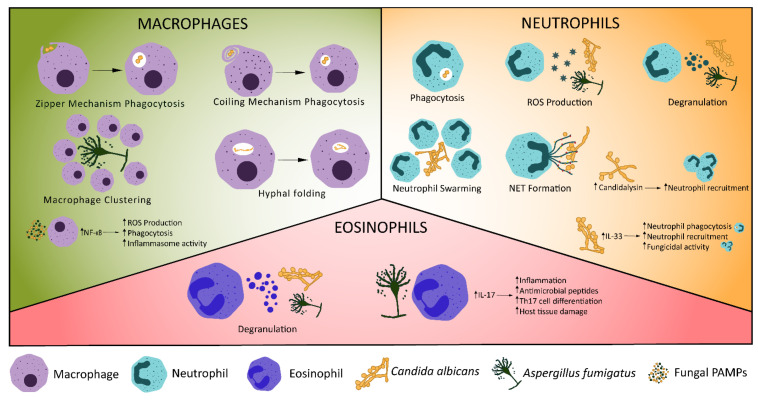
Cellular Innate Immune Control of Fungal Infections. Various mechanisms exist for the control of fungal infections by the innate immune system. Macrophages phagocytose fungi, undergo macrophage clustering or fold phagocytosed hyphae. Recognition of fungal ligands, such as candidalysin, stimulates production of IL-1β, triggering neutrophil recruitment [70]. Increased expression of IL-33 in *C. albicans* infection triggers neutrophil recruitment and phagocytosis [71]. Neutrophils may also release reactive oxygen species (ROS) or neutrophil extracellular traps, degranulate, phagocytose fungi or undergo swarming. Eosinophils have antifungal effects through degranulation [72] and production of IL-17, which stimulates pro-inflammatory signalling, production of antimicrobial peptides and Th17 cell differentiation [73,74].

### 2.4. Neutrophils

Neutrophil recruitment to fungi, like macrophages, is driven by detection of PAMP/DAMPs by PRRs (Table 1). Mincle binds α-mannose and other fungal cell wall components, resulting in pro-inflammatory signalling and recruitment of neutrophils [30,75]. RIG-I-like receptor detection of double stranded RNA stimulates a strong neutrophil-mediated antifungal response via MDA5/MAVS signalling [26,27]. Candidalysin, produced by hyphal *C. albicans*, appears to be a potent stimulator of innate immune responses in mucosal, central nervous system and systemic infections [70,76,77]. Candidalysin stimulates IL-1β production via a CARD9-dependent mechanism, which in turn leads to CXCL1-mediated recruitment of neutrophils [70]. TRAF1 (induced by the pro-inflammatory cytokine TNF) inhibits CXCL1 in *C. albicans* infection [78,79], suggesting a regulatory role for TRAF1 to prevent excess neutrophil recruitment and activation. Epidermal growth factor receptor (EGFR) may also be key for immune responses to candidalysin. Inhibition of EGFR in mouse models of oral candidiasis reduces IL-1β and CXCL1 [80,81], suggesting EGFR may be the initial receptor responsible for CXCL1-mediated neutrophil recruitment. Neutrophil recruitment and survival are also reduced by EGFR inhibition, providing strong evidence of the link between candidalysin, EGFR and CXCL1-mediated neutrophil recruitment. IL-33 is another key mediator of neutrophil recruitment [82] and IL-33 knockout mice have increased mortality in *C. albicans* infection. Based on in vitro primary cell models, IL-33 operates via IL-23 and GM-CSF to promote phagocytosis by neutrophils [71]. IL-33 also suppresses IL-10 expression, resulting in superior fungicidal activity by neutrophils in vitro. IL-10 expression has been associated with persistent *C. albicans* infection in other in vitro data [83], which may be due to reduced neutrophil activity. IL-23 has additional mechanisms of aiding antifungal immunity. IL-23 deficient mice have increased myeloid cell apoptosis, resulting in reduced survival in systemic *C. albicans* infection [84]. Interestingly, this occurred independent of IL-17 and was unique to fungal infections.

Neutrophils have been shown to coordinate their migration to sites of infection through a process called neutrophil swarming [85]. Swarming inhibits the growth of several fungal pathogens in vitro: *C. albicans*, *C. auris*, *Candida glabrata*, *C. neoformans* and *A. fumigatus* [86,87,88]. Swarms were smaller for yeast-locked *C. albicans*, *C. auris* and *C. glabrata* (which are unable to hyphate) compared to wild type *C. albicans* [87]. Furthermore, using an in vitro model of *A. fumigatus* infection, swarms appeared to preferentially form around hyphae [88]. This implies a potential correlation between hyphae formation and neutrophil swarming, which requires further investigation. Swarming in fungal infections is dependent on LTB4, meaning it operates by the same mechanism as swarming in other infections or injury [86,89,90].

Following migration to sites of infection, neutrophils have several mechanisms to eliminate fungal pathogens, including phagocytosis and degradation in the phagolysosome, degranulation, production of ROS and neutrophil extracellular trap (NET) release (Figure 1). Neutrophils produce granules containing a range of bactericidal and fungicidal effectors, including myeloperoxidase, cathepsins, defensins and lactoferrin [91]. The secretion of effectors in degranulation leads to fungal killing and is preferentially used in Candida infections with pseudo-hyphae [92]. Degranulation was dependent on CXCR1 in *C. albicans* infection in a mouse model, demonstrating a novel function of murine CXCR1 which correlates with evidence that human CXCR1 promotes oxidative and non-oxidative bactericidal activity by neutrophils [93,94]. Neutrophils also produce ROS, such as superoxide or hydrogen peroxide, which can be used intracellularly to kill phagocytosed fungi, or extracellularly to target hyphae [95,96]. Neutrophils are capable of expelling chromatin covered in antimicrobial proteins, leading to entrapment and killing of extracellular pathogens [97]. These neutrophil extracellular traps (NETs; Figure 1) are host protective against *C. albicans*, *Cryptococcus neoformans* and *Aspergillus nidulans* in in vitro human neutrophil models [87,97,98,99] and may be involved in swarm initiation [87,100]. NETs have also been demonstrated to directly stimulate Th17 cell differentiation, via TLR2 and RORγt, corresponding with increasing IL-17 and GM-CSF production [101]. IL-17 and GM-CSF both stimulate neutrophil activity in fungal infection [71,102], creating a feedback loop, in which NETs promote an adaptive immune response and additional neutrophil activity. A subpopulation of neutrophils also produce IL-17, which may further feed into this positive feedback loop [103]. Recent evidence suggests NETosis induced by *C. albicans* can occur independent of Peptidylarginine deiminase 4 (PAD4), contradicting established literature that NETosis is PAD4-dependent [104,105,106,107]. Alternative studies demonstrated PAD4 is not necessary for NET formation or neutrophil-mediated control of *A. fumigatus* infection [108,109]. Further research is needed to clarify whether PAD4-independent NETosis is a non-canonical mechanism of NETosis for all stimuli or is a phenomenon unique to fungal infections. Despite their fungicidal role, NETs may have an overall detrimental effect on the host. NET proteins intended to eliminate fungi have been observed bound to, but not killing, *C. albicans* and inducing apoptosis of host cells [110]. Furthermore, inhibition of NETosis reduced *A. fumigatus* burden in mouse models of invasive pulmonary aspergillosis [111]. However, these experiments inhibited NETosis by generating PAD4 knockout mice, which does not account for the possibility of PAD4-independent NETosis. Alternative methods of NETosis inhibition may be required to provide greater support to these conclusions.

Transfer of phagocytosed *A. fumigatus* conidia from neutrophils to macrophages by a β-glucan dependent mechanism has been observed in zebrafish [112]. The function of shuttling, and significance in fungal clearance, remains unclear, with conflicting arguments stating shuttling is a fungal strategy to avoid degradation by neutrophils or that shuttling is a host strategy to facilitate antigen presentation and initiate adaptive immune responses [38,112].

### 2.5. Other Innate Immune Components of Fungal Protection

Though typically associated with allergic disease, eosinophils can detect and respond to fungal infection, primarily through degranulation [72,113]. In a mouse model of acute *A. fumigatus* infection, eosinophils expressed RORγt, IL-23R and IL-17, which all have pro-inflammatory functions [73]. Increased IL-17 expression also occurred independent of IL-23 signalling, suggesting *A. fumigatus* promotes IL-17 production by eosinophils by an unknown mechanism. The pro-inflammatory IL-17 phenotype helps to protects against *A. fumigatus* infection (Figure 1), but is also responsible for tissue damage and lung pathology [73,74]. Eosinophils play a dichotomous role in fungal infection, whereby they both protect and damage the host.

Innate lymphoid cells (ILCs) are a rapidly emerging area of research. In a mouse model of oropharyngeal candidiasis, ILCs in the oral mucosa were the primary source of pro-inflammatory IL-17 during *C. albicans* infection, acting as the first line of defence in the antifungal response. Depletion of ILCs increased susceptibility to *C. albicans* infection, with increased fungal burden and greater reduction in body weight [114]. 3 distinct ILC subpopulations have been identified. Type 3 ILCs (ILC3s) in acute *A. fumigatus* lung infection in mice produce IL-22, a critical cytokine for clearance of *A. fumigatus* [115]. Stimulation of ILC3s by cytokines in vitro or *Citrobacter rodentium* infection in vivo resulted in stabilisation of HIF-1α, inducing glycolysis and RORγt production. Activation of ILC3s, characterised by secretion of IL-17 and IL-22, is also dependent on production of mitochondrial ROS, which aids HIF-1α and RORγt stabilisation [116]. Validation of this mechanism following stimulation with fungal pathogens is required.

Complement has previously been demonstrated to be crucial in responses to fungal infection, with complement-deficient mouse and guinea pig models being more vulnerable to *Candida* infections [117,118]. CD11b (also called CR3) is a common β subunit of β2 integrin complement receptors. While survival is not affected, CD11b knockout mice have reduced pro-inflammatory cytokines and reduced phagocytic activity by neutrophils during *A. fumigatus* infection but enhanced neutrophil infiltration [119]. Hence, complement activation of CD11b may be vital for triggering phagocytosis but plays no role in neutrophil recruitment. CR3 can also bind to β-glucan, triggering phagocytosis of β-glucan-bearing particles [120,121]. Binding of CR3 to β-glucan in *C. albicans* infection initiates a complex, temporally regulated pathway that can differentially upregulate neutrophil swarming and NETosis [122]. Inhibition of CR3 during in vitro *A. fumigatus* infection reduced production of IL-8 and MCP-1 and reduced activation of NF-κB, demonstrating CR3 has a role in mediating release of pro-inflammatory cytokines in *A. fumigatus* infection [123]. Together these studies reveal CR3 to be a complex, multifaceted protein, with important antifungal roles beyond the complement cascade.

## 3. Failures of Innate Immunity in Disease

### 3.1. Candida albicans

While usually a commensal fungus, commonly colonising the gastrointestinal and genitourinary tracts of most humans, *C. albicans* is also an opportunistic pathogen [124,125]. Immunocompromised patients are most at risk of candidiasis, whether their immune defects are caused by HIV/AIDS, immunosuppressive drugs, old age or genetic disorders [126]. Chronic mucocutaneous candidiasis (CMC) is recurrent or persistent infections of mucosal membranes caused by *Candida* species, primarily by *C. albicans* [127]. Severity of CMC is highly variable and dependent on the anatomical location of *C. albicans* infection: from soreness around the mouth and ulceration in oral cavity infections to abdominal pain and severe diarrhoea in digestive tract infections [128]. Although progression of CMC to systemic candidiasis is relatively rare, acquisition of invasive candidiasis in intensive care units is fairly common (7.07 episodes per 1000 admissions) and has a 42% 30-day mortality [129].

Susceptibility to recurrent *Candida* infection in CMC is caused by various host genetic mutations, the most common being STAT1 gain of function mutations [130]. These mutations increase STAT1 responses to IFNα, IFNβ, IFNγ and IL-27, causing repressed development of IL-17 T cells and susceptibility to mucosal *Candida* infection [131]. An alternative genetic mutation underlying CMC is autosomal recessive CARD9 deficiency [132]. CARD9 is an adaptor protein utilised by a variety of CLRs, such as Dectin-1, to stimulate NF-κB signalling. CARD9 deficiency has been demonstrated to cause reduced cytokine production by human patient peripheral blood mononuclear cells and impaired neutrophil recruitment in in vivo mouse models, leading to increased susceptibility to systemic candidiasis [133,134]. While several mutant CARD9 alleles associated with CARD9 deficiency have been revealed [132], less established is the effect other gene mutations may have on CARD9 expression. Ovarian tumour deubiquitinase family member 1 (OTUD1) has recently been identified as a positive regulator of CARD9. OTUD1 deubiquitinates ubiquitinated CARD9, leading to CARD9 activation. Infection of OTUD1 homozygous knockout mice with *C. albicans* led to reduced mouse survival and increased fungal burden in kidney, lung and spleen slices compared to wild type mice [135]. OTUD1 mutations may cause susceptibility to *C. albicans* infection through CARD9 deficiency (Figure 2), highlighting a novel cause of CMC and introducing a potential therapeutic target for treatment of CARD9 deficiency.

Given its key role in NF-κB signalling, MyD88 defects have been associated with susceptibility to a range of infectious diseases [136]. MyD88 knockout mice are highly susceptible to *C. albicans* infection, with increased mortality and fungal burden [137]. In a zebrafish wound model, MyD88 deficiency caused reduced recruitment of both local and distant neutrophils, which was maintained over 6 hours [138]. MyD88 deficiency, however, did not impair neutrophil activation, which was dependent on MAVS signalling. Although only studied in a wound model, if this mechanism is maintained in fungal infection, it would suggest patients with MyD88 deficiency have impaired neutrophil recruitment (but no defects in neutrophil activation), which causes increased susceptibility to *C. albicans* infection (Figure 2).

IL-17 is a key regulator of antifungal immunity. IL-17 knockout mice have high susceptibility to *C. albicans* infection and reduced levels of neutrophil-recruiting chemokines [139,140]. A strong association has been demonstrated between candidiasis and use of IL-17 inhibitors (used in treatment of several inflammatory diseases), with significant downregulation of 9 pro-inflammatory cytokines or neutrophil-recruiting chemokines [141]. A recent clinical study demonstrated 16 CMC patients had lower serum IL-17 levels than healthy controls [142], supporting previous evidence that IL-17 is impaired in CMC patients [143]. IL-17 defects have been attributed to STAT1 gain of function mutations, IL-17 mutations and anti-IL-17 autoantibodies [143,144,145,146].

**Figure 2 jof-08-00805-f002:**
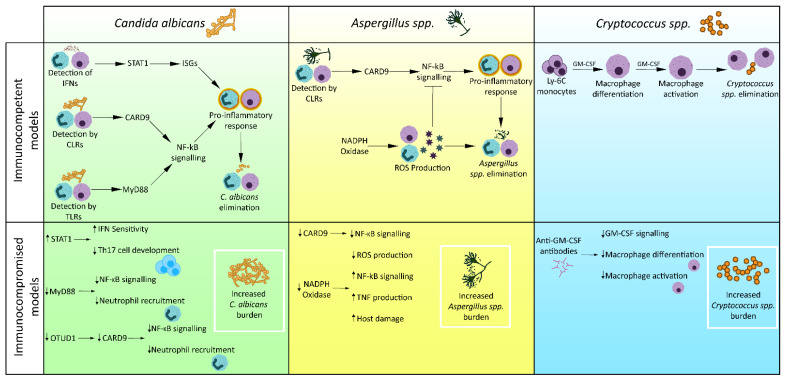
Failures of innate immunity in fungal infection. Examples of specific failures of innate immunity which lead to susceptibility to fungal infection. STAT1 gain of function mutations increase sensitivity to IFNs, leading to increased susceptibility to fungal infections [131]. OTUD1 or MyD88 deficiencies result in reduced transcription of NF-κB, resulting in reduced inflammatory response and increased *C. albicans* burden [135,138]. CARD9 deficiency causes reduced NF-κB transcription, resulting in reduced inflammatory response and increased *A. fumigatus* burden [147]. NADPH oxidase deficiency in CGD patients reduced production of ROS, reducing the ability to kill *A. fumigatus* and removing ROS-mediated inhibition of NF-κB, resulting in excess TNF production and host injury [41,131]. Anti-GM-CSF antibodies prevent macrophage differentiation and activation, resulting in inability to control *C. neoformans* infection [148].

### 3.2. Cryptococcus spp.

Cryptococcosis, caused by *Cryptococcus* spp., is a global issue, causing roughly 181,000 deaths annually worldwide and being responsible for 15% of all AIDS-related deaths [149,150]. Risk factors, other than HIV infection, include diabetes mellitus, cirrhosis, blood disorders or prolonged immunosuppression [131]. Cryptococcal infections typically begin in the lungs then disseminate to cause infections in the central nervous system, either as free yeasts or inside ‘trojan horse’ phagocytic cells [151]. *Cryptococcus neoformans* is the primary cause of cryptococcosis but *Cryptococcus gattii* is a rapidly emerging species, capable of causing infection in immunocompetent individuals, with a serious outbreak in the 2000s in Vancouver Island, Canada [152,153]. Unlike other pathogenic fungi, *Cryptococcus* spp. are yeast-locked and do not form hyphae during infection [154].

A number of patients with cryptococcosis have presented with anti-GM-CSF autoantibodies [155,156,157]. GM-CSF autoantibodies have previously been linked to pulmonary alveolar proteinosis, a rare condition affecting the ability of alveolar macrophages to remove excess surfactant, demonstrating an association between GM-CSF deficiency and macrophage function [147,158]. More recent investigations show impaired activation of macrophages in GM-CSF knockout mice with cryptococcal lung infection. Furthermore, GM-CSF deficiency in these mice reduced the differentiation and maturation of Ly-6C monocytes (macrophage precursor cells) into mature alveolar macrophages [148]. Therefore, cryptococcosis in patients with anti-GM-CSF autoantibodies may be caused by reduced levels of mature alveolar macrophages and an inability to activate these macrophages (Figure 2).

### 3.3. Aspergillus spp.

Infections caused by *Aspergillus* spp. most commonly occur in the lower respiratory tract or lungs [159]. *A. fumigatus* is the second most common cause of fungal infections in hospitalised patients, behind *C. albicans* [160]. *Aspergillus* infection is responsible for a range of severe pulmonary diseases, including: invasive pulmonary aspergillosis, chronic pulmonary aspergillosis and allergic bronchopulmonary aspergillosis [160]. Besides causing irreversible lung damage, these conditions can also lead to systemic aspergillosis.

Stem cell transplants (SCTs; used in treatment of some cancers and blood disorders) are a major risk factor for *Aspergillus* infection. 9.3% of SCT recipients develop invasive fungal disease, most frequently caused by *Aspergillus* spp., resulting in 70.8% mortality 1 year post diagnosis of fungal infection [161]. Following a SCT, patients are profoundly cytopenic while the immune system recovers and often receive additional immune suppression to prevent graft vs. host disease [162]. This cytopenia leads to the extreme vulnerability to fungal infection.

*Aspergillus* infection is particularly common in patients with chronic granulomatous disease (CGD). CGD results from defective NADPH oxidase complexes in phagocytes, leading to an inability to produce ROS [131]. Consequently, CGD patients are less able to kill pathogens so are highly susceptible to *Aspergillus* infection. Cagnina et al., recently showed excess TNF production in a CGD mouse model challenged with *Aspergillus* triggers additional neutrophil recruitment, driving a pro-inflammatory spiral which is responsible for host lung damage [41]. This supports other research demonstrating exaggerated TNF responses are responsible for host injury [163,164]. ROS are believed to inhibit NF-κB signalling to prevent hyper-inflammation [165,166]. Inability to produce ROS in CGD patients eliminates ROS-mediated NF-κB inhibition, resulting in a hyperinflammatory environment and host injury (Figure 2).

Although mainly associated with candidiasis, CARD9 deficiency is increasingly being linked to susceptibility to a wider range of fungal infections, including *Aspergillus* [132,167]. CARD9 deficiency caused far greater increases in mortality in mice during C. albicans infection compared to *A. fumigatus* infection [134,168], suggesting CARD9 exhibits some redundancy in control of *Aspergillus*.

## 4. Host-Directed Therapies

There are many issues concerning current antifungal treatment: toxicity of antifungals, low effectiveness, slow development of new antifungals and the rise of antifungal resistance [169]. A potential alternative strategy to treating infectious disease is therapeutic targeting of the immune system, an approach that should be more resilient against emerging drug resistance. Host-directed therapies (HDTs) stimulate host cellular pathways and activate immune responses to aid clearance of pathogens [170]. HDTs have been proposed as an adjunctive therapy alongside current antifungals [171].

Some HDTs function by promoting increased immune cell production or recruitment, to bolster the innate immune response. G-CSF is an endogenous signalling molecule that induces formation of granulocyte precursor cells, which will later differentiate into mature neutrophils. GM-CSF has a similar role in immune cell development, but with a wider spectrum of activity [172,173]. G-CSF/GM-CSF treatment aids clearance of *C. albicans* and *A. fumigatus* in in vitro models and in vivo rabbit and mouse models [174,175,176]. In two separate cases of relapsing C. *albicans* meningoencephalitis in CARD9-deficient patients, treatment with either G-CSF or GM-CSF resulted in complete clinical remission [177,178]. In another case, a paediatric patient with a history of CMC and CARD9 deficiency was successfully treated for invasive *C. albicans* infection by a combination of G-CSF and antifungals [179]. The primary mechanism believed to underly G-CSF/GM-CSF treatment is increased production, maturation, proliferation and activation of neutrophils, macrophages, monocytes and eosinophils [180]. However, G-CSF/GM-CSF have also been suggested to modulate immune activity by stimulating pro-inflammatory cytokine production, phagocytosis and ROS production, though this is based on in vitro evidence and may not be observed in human fungal infections [181,182,183,184]. Therefore, G-CSF/GM-CSF shows potential for treating *C. albicans* and *A. fumigatus* infections in immunodeficient patients. Targeting spleen tyrosine kinase (syk) was suggested as a potential HDT after syk was shown to be critical for protection against C. *albicans* infection in mice models, through regulation of neutrophil responses [185,186]. Syk is a downstream signalling molecule of several fungal PRRs, such as Dectin-1 [185,187], meaning syk is critical for the initial recognition of fungal infection, as well as neutrophil swarming, phagocytosis, NETosis and ROS production. However, while syk inhibitors have been developed and approved for clinical use [188], to the best of our knowledge, there are currently no pharmacological syk stimulators for experimental or clinical use, introducing significant barriers to further investigation of syk as a potential HDT. Rhesus theta defensin-1 (RTD-1) is an antimicrobial peptide, with potent antifungal properties in both in vitro and in vivo mouse models. RTD-1 also promotes neutrophil recruitment and reduces TNF, IL-1β and IL-17 production in *C. albicans* infected mice [189,190]. Furthermore, RTD-1 suppresses pro-inflammatory cytokines in in vitro and in vivo mouse models, reducing host damage, improving long term outcomes and improving pathogen clearance [191,192,193,194]. Hence, RTD-1 represents a promising new class of therapy, capable of modulating host responses to improve long term outcomes to infection, while also having a direct antifungal effect.

HDTs may also modulate innate immune cell responses, to improve their fungicidal activity. Following successful clinical trials, interferon gamma (IFNγ) is already used as a HDT as an antifungal prophylaxis and halves the occurrence of acute *Aspergillus* infection in CGD patients [195,196,197]. IFNγ stimulates increased ROS production by granulocytes, as well as promoting Th1 responses and enhanced macrophage activity [198,199]. Recent evidence has suggested IFNγ production is impaired in patients with chronic pulmonary aspergillosis, suggesting IFNγ therapy may also be beneficial in these patients [200]. Another potential HDT is HIF-1α (Hypoxia Inducible Factor) stabilisation. Hif-1α stabilisation is protective against *Mycobacterium marinum* infection in zebrafish larvae via IL-1β, which stimulates antimicrobial nitric oxide production [201,202]. In addition, Hif-1α deficient mice have been shown to be more susceptible to *C. albicans* infection [203]. Hif-1α stabilisation has potential as a HDT for treatment of fungal infections but requires appropriate in vitro models that allow both genotypic and phenotypic characterisation of the effects of Hif-1α stabilisation on fungal infection and immune cell behaviour.

An emerging HDT opportunity in fungal diseases is an increased understanding of trained innate immunity, where innate immune cells exhibit long-term adaptive characteristics after immune challenge [204]. Much of the research in this area has focused on training immunity with bacterial products, such as lipopolysaccharide (LPS) or bacillus Calmette-Guérin vaccine (BCG) that can trigger different trained immunity programmes that protect against subsequent infections [205]. However, some of the earliest demonstrated examples of trained immunity were in murine studies using low-dose *Candida* spp infection in T- and B-cell depleted animals, that showed protection against a subsequent lethal dose of fungal infection [206]. The fungal cell wall component β-glucan was sufficient to provide trained innate immunity protection and this response was dependent on functional circulating monocytes [206]. There is emerging evidence that challenge with avirulent *Candida* spp can provide protection against sepsis caused by bloodstream fungal infections, with a potential role for GR-1^+^ putative myeloid-derived suppressor cells (MDSCs) [207,208]. These findings open up the possibility of using trained immunity to treat fungal infections, but there is more to understand on the exact mechanisms, programmes and specificity of fungal-induced trained immunity before potential exploitation in the clinic [209].

## 5. Conclusions

From established processes, such as phagocytosis and production of ROS, to recently discovered phenomena, such as hyphal folding, the innate immune system is critical in host immune responses to fungal infections. The fact that new mechanisms are still being uncovered highlights the complexity of the innate response. However, many questions remain unanswered. It remains to be seen how clinically significant recent observations from in vitro/ex vivo experiments are, for example, NET release, fungal shuttling and hyphal folding. The ability to unravel these questions relies on in vivo models for fungal infections, which are translatable to human disease and immune responses. These models are also vital to understand mechanisms underlying increased vulnerability to fungal infections in immunocompromised individuals and the effects of HDTs on fungal clearance and immune cell behaviour.

## Figures and Tables

**Table 1 jof-08-00805-t001:** Pattern recognition receptors in fungal infection.

Pattern Recognition Receptor	Localisation	Cell Expression	Adaptor Proteins	Effectors	Pathogen-/Damage-Associated Molecular Patterns Recognised	Fungal Species	References
TLR2	Plasma membrane	Monocytes, macrophages, dendritic cells, mast cells, neutrophils	MyD88, Mal	NF-κB, TNF, TGFβ, IL-10, IL-12, IFNγ	Phospholipomannan, β-glucans	*C. albicans, A. fumigatus, P. brasiliensis*	[28,29,30,31]
TLR4	Plasma membrane, endosome membrane	Monocytes, macrophages, dendritic cells, mast cells, neutrophils, B cells, intestinal epithelium	MyD88, Mal, TRIF, TRAM	NF-κB, TNF, IL-8, Type I IFN	O-linked mannosyl, Mannan, Glucuronoxylomannan	*C. albican* *s, A. fumigatus*	[28,29,30,31]
TLR7	Endosome membrane	Monocytes, macrophages, dendritic cells, B cells	MyD88	IFN-β, Type I IFN	ssRNA	*C. albicans*	[28,30,31,32]
TLR9	Endosome membrane	Monocytes, macrophages, dendritic cells, B cells	MyD88	NF-κB, IL-12, TNFα	Unmethylated DNA with CpG motif	*Candida spp., C. neoformans, A. fumigatus, P. brasiliensis, M. furfur*	[28,30,31,33,34]
Dectin-1	Plasma membrane	Monocytes, macrophages, dendritic cells, neutrophils, mast cells, some T cells	hemITAM	IL-2, IL-6, IL-10, IL-23	β-1,3-glucans	*Candida spp., C. neoformans, A. fumigatus, H. capsulatum, S. cerevisiae, P. brasiliensis*	[28,30,31]
Dectin-2	Plasma membrane	Monocytes, macrophages, dendritic cells, neutrophils	ITAM-FcRγ	TNFα	Mannose	*C. albicans, C. glabrata, C. neoformans, A. fumigatus, H. capsulatum*	[28,30,31]
Mincle	Plasma membrane	Monocytes, macrophages, dendritic cells, neutrophils, mast cells, some B cells	ITAM-FcRγ	NF-κB, IL-1, IL-6, IL-10 IL-12, IL-23	α-mannose, glyceroglycolipid, mannosyl fatty acids, MSG/gpA	*A. fumigatus, C. albicans, P. carinii, Malassezia spp.*	[30,31,35]
DC-SIGN	Plasma membrane	Macrophages, dendritic cells, activated B cells	LSP1	IL-10	Mannose, N-linked mannans, galactomannans	*C. albicans, C. neoformans, A. fumigatus, S. cerevisiae*	[28,30,31]
Mannose Receptor	Plasma membrane	Macrophages, Kupffer cells, endothelial cells	Associated with FcRγ and GBR2, exact mechanism unknown	TNF, IL-1β	Mannose, α-glucans, chitin	*C. albicans, C. neoformans, A. fumigatus, H. capsulatum, S. cerevisiae, P. brasiliensis*	[28,30,36,37]
MDA5	Cytoplasm	Monocytes, macrophages, dendritic cells, B cells, epithelial cells, endothelial cells, fibroblasts	CARDs, MAVs	NF-κB, Type I IFN, Type III IFN, TNFα, IL-12,	dsRNA	*C. albicans, A. fumigatus*	[26,30]

## Data Availability

Not applicable.

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
