# Peer review of "A Fun-Guide to Innate Immune Responses to Fungal Infections"

_jof, 2022, doi:10.3390/jof8080805_

Round 1

Reviewer 1 Report

The present manuscript sought to review the innate immune response against the 3 main fungal pathogens. The information is well supported by up-to-date scientific literature. The figures are great. I only suggest that figure 2 could be made larger (perhaps a full page) for easier viewing and better resolution.

Author Response

We thank the reviewer for their positive comments and the time taken to review this manuscript. We agree that figure 2 could be made larger. We will upload a high resolution image in order that the editing team at JoF have the means to do this in the published manuscript. 

Reviewer 2 Report

This review examines three major fungal pathogens with respect to clinical challenges, innate immunology, and application to therapeutic strategies.  First, the burden of infections by Candida, Cryptococcus and Aspergillus is considered alongside the shortcomings of research commitments and pharmaceutic development, relative to the scope of the medical mycological problem.  Second, the cellular and molecular immunology of innate response to fungal pathogens is broadly reviewed.  Third, consideration is given to the ways that human host defense fails, resulting in various fungal infections. Fourth, several promising avenues for leveraging the foregoing knowledge to produce host directed therapies are critically discussed.  The review is quite thorough and comprehensive within these areas. Depth and breadth are appropriate. The article is well written and enjoyable to read.

Review articles that address a particular gap in knowledge are especially beneficial to their fields.  To be most effective, the gap in knowledge needs to be clear from the start.  I infer that the authors were trying to address the current gap that exists between a relatively large body of information about mechanisms innate immune host defense and the implementation of strategies that can help to combat infection in patients.  If this is so, the gap could be stated more forcefully in the abstract and/or introduction, and I believe that would be beneficial to appropriately orienting the reader to the goal of the review.

Detailed comments in order:

Ln112-113. The sentence seems to state that Fc receptors bind to complement proteins, which is incorrect.  The cited reference does not support this, and in fact states Fc receptors bind to particles opsonized by antibody.  The authors probably meant to say Fc R bind to “opsonizing IgG” or perhaps they meant to refer to complement receptor instead of Fc receptor.

Ln114, 134 and surrounding paragraph. In a nice discussion of innate immune mechanisms of cellular interaction with fungal particles, phagocytosis and even more specialized areas like dragotcytosis and coiling phagocytosis are mentioned. It may be consistent with the degree of detail being pursued to also mention literature on vomocytosis of fungal particles and its possible significance.

Ln 186-187.  Discussion of TRAF1 as a negative regulator of neutrophil recruitment/activation could benefit from a little more context on what is likely to be triggering TRAF1 in a relevant scenario. 

Ln 203ff, 387.  The authors have nice, separate discussions about the roles of macrophage clustering and neutrophil swarming in response to fungal infection.  I am curious if the authors are aware of literature concerning the interaction of these two responses in the context of fungal infection. In a sterile injury model, Uderhardt, et al (Cell 2019, 177:541) showed that macrophages “cloak” tissue damage to regulate neutrophil recruitment, which often is associated with tissue damage. Admittedly, Uderhardt is a different model of tissue pathology, but it may connect with Cagnina, et al, which they reference (ln 387) in connection with Aspergillus host defense.  Failure of macrophages to adequately cloak tissue damage in fungal invasion may play into the tissue damage noted in Cagnina, et al. Readers might find that to be a helpful connection between host defense against invasive fungi and innate responses in other models, as they authors have similarly done with the effects of MyD88 in a wound healing model (ln 317ff).

Ln 278. The role of complement recognition by CR3 is discussed. It seems appropriate to also mention work by Reichner and others indicating that CR3 can also interact directly with beta-glucan.  

Author Response

This review examines three major fungal pathogens with respect to clinical challenges, innate immunology, and application to therapeutic strategies.  First, the burden of infections by Candida, Cryptococcus and Aspergillus is considered alongside the shortcomings of research commitments and pharmaceutic development, relative to the scope of the medical mycological problem.  Second, the cellular and molecular immunology of innate response to fungal pathogens is broadly reviewed.  Third, consideration is given to the ways that human host defense fails, resulting in various fungal infections. Fourth, several promising avenues for leveraging the foregoing knowledge to produce host directed therapies are critically discussed.  The review is quite thorough and comprehensive within these areas. Depth and breadth are appropriate. The article is well written and enjoyable to read.

Review articles that address a particular gap in knowledge are especially beneficial to their fields.  To be most effective, the gap in knowledge needs to be clear from the start.  I infer that the authors were trying to address the current gap that exists between a relatively large body of information about mechanisms innate immune host defense and the implementation of strategies that can help to combat infection in patients.  If this is so, the gap could be stated more forcefully in the abstract and/or introduction, and I believe that would be beneficial to appropriately orienting the reader to the goal of the review.

RESPONSE: We thank the reviewer for their positive comments and the time taken to review this manuscript. The authors agree with the reviewer that specifying the gap in knowledge we aim to address would be helpful for readers. We have added sentences in the abstract (line 15-17) and introduction (line 64-66) to make the gap in knowledge more clear.

Detailed comments in order:

Ln112-113. The sentence seems to state that Fc receptors bind to complement proteins, which is incorrect.  The cited reference does not support this, and in fact states Fc receptors bind to particles opsonized by antibody.  The authors probably meant to say Fc R bind to “opsonizing IgG” or perhaps they meant to refer to complement receptor instead of Fc receptor.

RESPONSE: Apologies for this error. The sentence has been changed to say Fc receptors bind to “opsonising antibodies” (line 123).

Ln114, 134 and surrounding paragraph. In a nice discussion of innate immune mechanisms of cellular interaction with fungal particles, phagocytosis and even more specialized areas like dragotcytosis and coiling phagocytosis are mentioned. It may be consistent with the degree of detail being pursued to also mention literature on vomocytosis of fungal particles and its possible significance.

RESPONSE: The authors agree with the reviewer that vomocytosis would fit well in this paragraph. Sentences have been added introducing vomocytosis and potential links with dragotcytosis, with appropriate references (line 151-158).

Ln 186-187.  Discussion of TRAF1 as a negative regulator of neutrophil recruitment/activation could benefit from a little more context on what is likely to be triggering TRAF1 in a relevant scenario.

RESPONSE: A brief statement has been added explaining TRAF1 expression is induced by the pro-inflammatory cytokine TNF, with an appropriate reference (line 209-210).

Ln 203ff, 387.  The authors have nice, separate discussions about the roles of macrophage clustering and neutrophil swarming in response to fungal infection.  I am curious if the authors are aware of literature concerning the interaction of these two responses in the context of fungal infection. In a sterile injury model, Uderhardt, et al (Cell 2019, 177:541) showed that macrophages “cloak” tissue damage to regulate neutrophil recruitment, which often is associated with tissue damage. Admittedly, Uderhardt is a different model of tissue pathology, but it may connect with Cagnina, et al, which they reference (ln 387) in connection with Aspergillus host defense.  Failure of macrophages to adequately cloak tissue damage in fungal invasion may play into the tissue damage noted in Cagnina, et al. Readers might find that to be a helpful connection between host defense against invasive fungi and innate responses in other models, as they authors have similarly done with the effects of MyD88 in a wound healing model (ln 317ff).

RESPONSE: The authors thank the reviewer for their suggestion, this is really interesting work. Uderhardt et al. shows an interaction between macrophages and neutrophils in a sterile injury model. We have added sentences describing the possible association between macrophage clustering and cloaking, and how lack of cloaking could permit the excess tissue damage noted by Cagnina et al. (line 112-121).

Ln 278. The role of complement recognition by CR3 is discussed. It seems appropriate to also mention work by Reichner and others indicating that CR3 can also interact directly with beta-glucan.

RESPONSE: We have added several sentences referring to Reichner’s and others’ work on CR3 and beta-glucan (line 307-314).

Reviewer 3 Report

Generally speaking, this is a well written and comprehensive review of both important background information and recent findings relating to the role of innate immunity in the response to fungal infections.  As host-directed therapies represent an important and somewhat conceptually novel approach to mitigating fungal infections, the section devoted to this topic is welcome.  To strengthen it, it would be worth calling attention to some recent observations regarding the engagement of trained immunity/trained innate immunity.  For example, see Lilly et al (PMID 34663098) for the induction of trained immunity and the potential role of Ly6G+ putative G-MDSCs in the protection against C. albicans bloodstream infection.